# Design of low-carbon planning model for vehicle path based on adaptive multi-strategy ant colony optimization algorithm

Qi Guo[1,*], Rui Li[2,*], Changjiang Zheng[3] and Gwanggil Jeon[4]

[1] Department of Automotive Engineering, Anhui Institute of Automotive Technology, Hefei, Anhui, China
[2] Business School, Wuxi Taihu College, Wuxi, Jiangsu, China
[3] College of Civil Engineering and Transportation, Hohai University, Nanjing, Jiangsu, China
[4] Department of Embedded Systems Engineering, Incheon National University, Incheon, Republic of South Korea
* These authors contributed equally to this work.



Corresponding author
Rui Li, 13251966170@163.com

## ABSTRACT

In contemporary transportation systems, the imperatives of route planning and optimization have become increasingly critical due to vehicles' burgeoning number and complexity. This includes various vehicle types, such as electric and autonomous vehicles, each with specific needs. Additionally, varying speeds and operational requirements further complicate the process, demanding more sophisticated planning solutions. These systems frequently confront myriad challenges, including traffic congestion, intricate routes, and substantial energy consumption, which collectively undermine transportation efficiency, escalate energy usage, and contribute to environmental pollution. Hence, strategically planning and optimizing routes within complex traffic milieus are paramount to enhancing transportation efficacy and achieving low-carbon and environmentally sustainable objectives. This article proposes a vehicle path low-carbon planning model, Adaptive Cooperative Graph Neural Network (ACGNN), predicated on an adaptive multi-strategy ant colony optimization algorithm, addressing the vehicle path low-carbon planning conundrum. The proposed framework initially employs graph data from road networks and historical trajectories as model inputs, generating high-quality graph data through subgraph screening. Subsequently, a graph neural network (GNN) is utilized to optimize nodes and edges computationally. At the same time, the global search capability of the model is augmented *via* an ant colony optimization algorithm to ascertain the final optimized path. Experimental results demonstrate that ACGNN yields significant path planning outcomes on both public and custom-built datasets, surpassing the traditional Dijkstra's shortest path algorithm, random graph network (RGN), and conventional GNN methodologies. Moreover, comparative analyses of various optimization methods on the custom-built dataset reveal that the ant colony optimization algorithm markedly outperforms the simulated annealing algorithm (SA) and particle swarm optimization algorithm (PSO). The method offers an innovative technical approach to vehicle path planning and is instrumental in advancing low-carbon and environmentally sustainable goals while enhancing transportation efficiency.

# INTRODUCTION

Traffic route planning constitutes one of the fundamental challenges in contemporary transportation systems, a significance underscored by the ongoing urbanization and escalating transportation demands. Traditional methods, such as Dijkstra's shortest path and A* algorithms, yield effective solutions under certain conditions. However, as urban transportation networks grow increasingly complex and dynamic, these traditional approaches reveal computational efficiency and adaptability limitations. Recent advancements in intelligent transportation systems (ITS) have heralded new prospects for transportation route planning (*Hu et al., 2020*). ITS combines advanced information, data communication, and electronic sensing technology. This integration enables comprehensive perception and intelligent processing of traffic information. As a result, ITS provides more accurate and real-time support for traffic route planning (*Njoku et al., 2023*). In the context of global green and low-carbon development, transportation route planning must extend beyond driving time and distance considerations to encompass environmental factors such as vehicle fuel consumption and carbon emissions. The advent of low-carbon transportation route planning mitigates traffic congestion, enhances transportation efficiency, and significantly curtails greenhouse gas emissions, marking a pivotal contribution to environmental protection and sustainable development (*Lin et al., 2020*). Achieving the objectives of low-carbon environmental conservation through scientifically and judiciously planned routes has emerged as a critical research trajectory in the current transportation sector.

Deep learning and machine learning techniques have progressively emerged as potent tools for addressing traffic route planning challenges, leveraging their robust data processing and pattern recognition capabilities. The urban transportation network is constantly changing over time, and unexpected situations such as traffic flow, road closures, and accidents can affect the rationality of route selection. Secondly, real-time data processing is the key to path planning, and it is necessary to quickly respond to data from vehicles and sensors to ensure the real-time and accurate planning of the path. In addition, environmental uncertainties such as weather and changes in road conditions also increase the difficulty of path planning. These factors work together to make path planning a computational challenge and a complex system optimization problem that requires consideration of multiple dynamic variables. Deep learning methodologies adeptly handle complex traffic environments and fluctuating road conditions by constructing intricate neural network models that autonomously learn significant features and patterns from vast traffic datasets (*Huang, Xu & Weng, 2020*). Convolutional neural networks (CNNs), renowned for their exceptional performance in image recognition and processing, employ two-dimensional convolutional operations that have been adapted for feature extraction and path optimization within traffic networks. Recurrent neural networks (RNNs), particularly their variant long short-term memory networks (LSTM), excel in

managing time-series data, rendering them suitable for predicting traffic flow and travel times.

Furthermore, GNN, a recent advancement in deep learning, is particularly adept at processing graph-structured data. Traffic road networks inherently possess a graph structure, with nodes representing intersections and edges symbolizing roads. GNNs enhance the understanding of complex associations and traffic dynamics within road networks by executing convolution operations on graph data, thereby providing more precise inputs for route planning (*Liu et al., 2020*).

In addition to deep learning methodologies, meta-heuristic algorithms are pivotal in path planning and network parameter optimization. The ant colony optimization (ACO) algorithm, a quintessential meta-inspired algorithm, derives inspiration from the natural foraging behavior of ants and their collaborative use of pheromones. ACO emulates this process, leveraging the positive feedback mechanism of pheromones to reinforce the selection of high-quality paths progressively. This approach is characterized by a robust global search capability and adaptability, effectively circumventing the pitfalls of local optima. The simulated annealing (SA) algorithm emulates the physical annealing process, probabilistically accepting suboptimal solutions and gradually reducing the "temperature," making it well-suited for intricate optimization problems. Particle swarm optimization (PSO) algorithms expedite the approach to the optimal solution by mimicking the foraging behavior of bird flocks, utilizing information sharing and collaboration among particles. Integrating these meta-heuristic algorithms into transportation route planning markedly enhances optimization efficiency (*Pustokhina et al., 2021*). When combined with GNN, the ACO algorithm further augments the model's global search capacity and path optimization performance. In path planning, the carbon emissions of different vehicles critically impact environmental sustainability. Accurately identifying green paths and optimizing routes to reduce travel distance and carbon emission costs is essential for monitoring green carbon emissions. Consequently, this article proposes a novel adaptive multi-strategy path planning method utilizing the ACO algorithm in conjunction with GNN, with the following specific contributions:

1) A graph data structure was established based on the characteristics of path planning and data attributes. Utilizing feature selection for subgraph screening facilitated the generation of input data.
2) The GNN predicted the outputs of nodes and edges in the route planning process. OptimOptimizingth planning and analytical identification, thereby it achieved optimal path planning and analytical identification
3) Comparative analyses between public and self-built datasets demonstrated that the proposed Adaptive Cooperative Graph Neural Network (ACGNN) framework yields superior recognition results, underscoring the potential and advantages of the ant colony optimization algorithm in ablation experiments.

The remainder of this article is organized as follows: "Related Works" introduces related works on meta-heuristic algorithms and path planning. In "Methodology", the ACGNN

framework is established. "Experiment Result and Analysis" details the experiments and result analysis. Discussion is provided in "Discussion", followed by the conclusion.

# RELATED WORKS

## Optimization study based on meta-heuristic class algorithm

*Ajeil et al. (2020)* proposed an efficient path-planning technique for mobile robots operating in known environments, utilizing simulated annealing neural networks to describe obstacles directly. This method generates segmentally linear paths *Yang & Meng (2021)* introduced a genetic algorithm for global path planning of mobile robots, where candidate path solutions are represented as individuals in the genetic algorithm and evolved using evolutionary operators. In each generation, the genetic algorithm individuals undergo local path refinement to correct and enhance the encoded paths. *Yang & Meng (2023)* addressed the instability of traditional genetic algorithms by proposing a new population-based incremental learning algorithm for path planning, employing a node probabilistic model and an edge library to generate promising paths. *Xu (2022)* presented a path-planning method based on an adaptive multistate ant colony algorithm, which utilizes an adaptive state transfer strategy to balance the relative importance of pheromone strength and desirability. This method also employs a direction-determination approach to resolve deadlocks, g the algorithm's global search capability for goal planning and obstacle avoidance. *Sanchez-Ibanez, Pérez-del-Pulgar & García-Cerezo (2021)* improved the standard ant colony optimization, termed age-based ant colony optimization. This enhanced ant colony algorithm, implemented alongside grid-based modeling of static and dynamic environments, effectively addresses the path planning problem.

## Drones and trajectory planning

*Karaman & Frazzoli (2020)* proposed a trajectory generation method based on the concept of search within a discrete space closely related to the behavioral layer. This method involves selecting the target configuration of the vehicle's motion in the Frenet coordinate system, using polynomials to derive a set of navigable trajectories, and selecting the optimal trajectory as the predicted path. The safety of this path is then verified through collision detection. Extensive experiments have demonstrated the practicality of this approach in standardized road environments. Additionally, *Karaman & Frazzoli (2020)* proposed an improved RRT* algorithm, addressing computational complexity and the stability of feasible solutions, yielding significant optimization. *Mashayekhi et al. (2020)* introduced the bidirectional RRT algorithm (RRT-Connect), which extends the random tree from both the start and end points, substantially enhancing planning efficiency and real-time performance. *Yuan et al. (2019)* proposed a dynamic path planning method based on a gated recurrent unit-recurrent neural network model for unknown environments. This method utilizes deep neural networks combined with sensor inputs to generate a new control strategy, outputted to a physical model to control the robot's motion and avoid obstacles. *Wu et al. (2020)* developed a new deep neural network (DNN)-based approach for real-time online path planning in unknown, cluttered environments. *Bahar, Ghiasi & Bahar (2012)* proposed a grid roadmap-based path

planning method to reduce the time required for path planning. This method implements a fast path planning strategy using a suggested roadmap and an artificial neural network (ANN) corridor search, operable in static, dynamic, and invisible environments. *Bozek et al. (2020)* investigated the development of an intelligent control system for wheeled robots and proposed an artificial neural network-based path planning algorithm. The control system comprises two artificial neural networks: one specifies the position and size of obstacles, and the other processes received information, coordinates, and target point orientation to form a continuous trajectory for obstacle navigation.

The studies above demonstrate substantial progress in applying metaheuristic algorithms to path planning. Researchers have proposed various optimization algorithms, including ant colony optimization, genetic algorithms, and population incremental learning, significantly enhancing the efficiency and accuracy of path planning. Improved ant colony and genetic algorithms, for instance, address traditional algorithms' stability and global search capability issues by incorporating novel strategies and optimization steps. Concurrently, deep learning methods are gaining prominence in the path planning domain. Researchers have devised numerous real-time dynamic path-planning algorithms by integrating neural networks and deep learning techniques. Consequently, optimizing machine learning methods through metaheuristic algorithms remains essential for advancing path-planning research.

## METHODOLOGY

### Meta-inspired ant colony algorithm

ACO is a bio-inspired optimization algorithm derived from the foraging behavior of ants in nature. It primarily addresses combinatorial optimization problems, such as the Traveling Salesman Problem (TSP) and path planning. The ACO algorithm attains the global optimal solution by emulating the information exchange and collaborative behavior among individual ants. This algorithm excels in discovering optimal or near-optimal paths within complex, multi-objective environments, making it well-suited for path planning in both static and dynamic scenarios (*Padmanaban & Sathiyamoorthy, 2020*). Its path selection mechanism, based on heuristic information and pheromone concentration, enables the rapid identification of the optimal solution from among multiple potential paths. Firstly, parameter initialization is performed by setting the number of ants $m$, the initial value of pheromone $\tau_0$, the pheromone importance factor $\alpha$, the heuristic information importance factor $\beta$, the pheromone volatility coefficient $\rho$, and the maximum number of iterations $T$. The solution is then reconstructed, with each ant constructing the solution independently. At each step of path selection, the ants determine the transfer probability based on the pheromone concentration and heuristic information on the path. For ant $k$ the probability of transferring from node $i$ to node $j p_{ij}^k$ is defined as.

$$p_{ij}^k = \frac{[\tau_{ij}(t)]^\alpha \cdot [\eta_{ij}]^\beta}{\sum_{l \in \text{allowed\_nodes}} [\tau_{il}(t)]^\alpha \cdot [\eta_{il}]^\beta} \tag{1}$$

where $\tau_{ij}(t)$ is the pheromone concentration on the path $(i, j)$ and $\eta_{ij}$ is the heuristic information, usually the visibility of the path. Based on this pheromone update, the ant needs to update the pheromone on the path after completing the construction of a solution once. Pheromone updating includes pheromone volatilization and addition of new pheromone. For the pheromone update on the path $(i, j)$ the formula is as follows:

$$\tau_{ij}(t + 1) = (1 - \rho) \cdot \tau_{ij}(t) + \Delta\tau_{ij} \tag{2}$$

where $\rho$ is the pheromone volatilization coefficient $(0 < \rho < 1)$, which indicates the decay of pheromone over time. $\Delta\tau_{ij}$ is the amount of newly added pheromone, determined by the sum of pheromone left by all ants on the path.

$$\Delta\tau_{ij} = \sum_{k=1}^{m} \Delta\tau_{ij}^{k}. \tag{3}$$

For the ant $k$, the amount of pheromone it leaves on the path $(i, j)$ is as follows:

$$\Delta\tau_{ij}^{k} = \begin{cases} \dfrac{Q}{L_k}, & \text{if } k \to (i, j) \\ 0, & otherwise \end{cases} \tag{4}$$

where $Q$ is a constant, and $L_k$ is the total path length of the solution constructed by ant $kk \to (i, j)$ represents that ant k has passed through point (i, j). Repeating the above process of constructing the solution and updating the pheromone until a predetermined maximum number of iterations T is reached or a satisfactory solution completes the optimal path planning (*Toaza & Esztergár-Kiss, 2023*). Based on experience, we set the maximum number of iterations for the ant colony to 100 and set the cut-off condition for model iteration as the difference between the two losses being less than 0.05.

### Graph network path planning

GNN represents a class of neural network models adept at processing graph-structured data. GNNs have found extensive applications in domains such as social network analysis, chemical molecule modeling, and recommender systems by learning the representation of graph data through the relationships between nodes and edges. These networks capture a graph's topology and node features by propagating and aggregating information among the graph nodes, thereby enabling the deep learning of graph data (*Shi & Rajkumar, 2020*). A graph G = (V, E), consists of a set of nodes V and a set of edges E. Nodes are connected by edges, and each node and edge can contain feature information. The main goal of a graph network is to learn a representation (embedding) of each node in the graph that reflects the node's features and its position and relationship in the graph structure. For a typical graph network, it is necessary to first initialize the nodes, the initial feature representation of each node $v \in V$ is $h_v^{(0)}$, which can be the feature vector of the node. In graph networks, each iteration consists of two main steps, message passing and aggregation. Each node receives messages from its neighboring nodes. For node $v$ the received message in the first $k$ iteration is denoted as

$$m_v^{(k)} = \sum_{u \in N(v)} M\left(h_u^{(k-1)} h_v^{(k-1)}, e_{uv}\right) \tag{5}$$

where $N(v)$ denotes the set of neighboring nodes of node $v$, $M$ is the messaging function, $h_u^{(k-1)}$ and $h_v^{(k-1)}$ are the feature representations of node $u$ and node $v$ at round $k-1$ respectively, and $e_{uv}$ is the feature of the edge between node $u$ and node $v$. After the initial message passing, further message aggregation is required, *i.e.*, each node updates its own representation based on the received messages. The representation of node $v$ at round $k$ iteration is updated as

$$h_v^{(k)} = U\left(h_v^{(k-1)}, m_v^{(k)}\right) \tag{6}$$

where, $U$ is the aggregation function which updates the representation of the node based on the previous round of representation of the node and the received messages. After the message delivery and aggregation, we can then perform the corresponding output. After $K$ rounds of iteration, the final representation of the node $v$ is $h_v^{(K)}$. This representation encompasses the node's initial characteristics and the neighbor information aggregated over multiple rounds of message passing, thereby reflecting the node's comprehensive information within the graph structure. Considering the subsequent path planning problem involving multiple discrete points, this article employs the softmax activation function to predict the discrete trajectories.

$$\hat{y}_v = \text{softmax}\left(W \cdot h_v^{(K)}\right) \tag{7}$$

where W represents the corresponding weights. In this article we use the cross-entropy function for the loss analysis of the model, and its specific calculation process is shown in Eq. (8):

$$\text{Loss} = \begin{cases} -\alpha \cdot \log y', & y = 1 \\ -\beta \cdot \log(1 - y'), & y = 0 \end{cases} \tag{8}$$

where $y'$ is the output of the model, and $y$ is the sample label. $\alpha, \beta$ are the weights of positive and negative samples, respectively.

Before training the GNN, it is essential to create appropriate graph data. The process of creating the graph data in this article is illustrated in Fig. 1:

By preprocessing the data from the distance network and the historical traffic trajectories, the road network used for the path planning experiments in this article is filtered from the global map data based on the coordinate range of the trajectory dataset. This constitutes the initial step of subgraph screening. However, further screening of the connectivity subgraphs from the road networks is necessary. Sub-figure screening helps to mitigate the category imbalance problem and accelerates subsequent model training. When handling large datasets, GNNs excel at feature extraction and modeling complex relationships within graph structures; however, their computational complexity increases significantly with larger data volumes, leading to higher processing costs. This study's

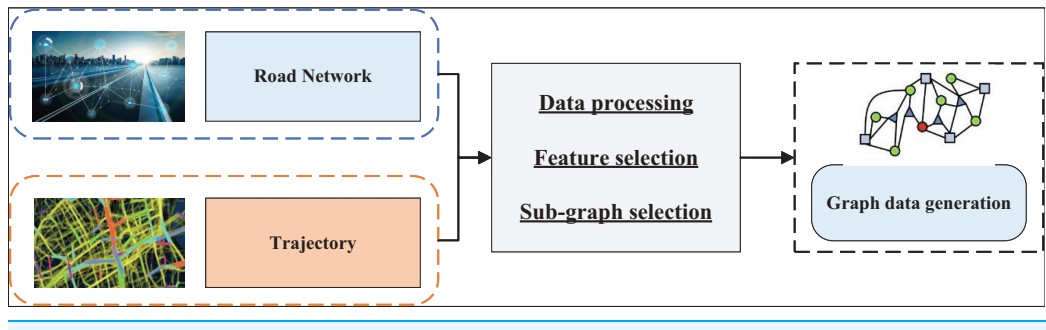

**Figure 1** **The process for the graph data generation.**

dataset is relatively small, reducing computational demands and enabling GNNs to operate efficiently for path-planning tasks. The smaller data scale decreases the time required for model training and inference and enhances resource utilization, ensuring real-time applicability and practicality in path optimization.

## The establishment of the ACGNN

Based on "Meta-inspired Ant Colony Algorithm" and "Graph network path planning", it is evident that the essence of the path planning problem is to find the shortest or most efficient path according to the users' preferences. Using graph data is particularly effective in accomplishing the corresponding route planning task. This article proposes a route planning network, ACGNN, which leverages the generation of graph data and GNN. The overall structure of this network is illustrated in Fig. 2:

Figure 2 illustrates the data processing process of the entire path optimization framework. The network first uses the corresponding graph data from the road network and historical trajectories. After selecting the required graph data based on subgraphs, this article uses the generated graph data as input to the model. Following the generation of the necessary graph data through subgraph filtering, this graph data is used as input for the model. The GNN serves as the computational network for optimizing the nodes and edges, and the optimization is carried out using the ant colony algorithm to obtain the final optimized output.

Consequently, the recommended nodes and edges are computed to generate the optimized paths. The combination of the ant colony algorithm and GNN is mainly reflected in the feature extraction of graph structure and the optimization of the path search strategy. Firstly, the node and edge features of the path graph are extracted using GNN to construct a spatial representation and generate preliminary path planning. Then, based on the graph structure information output by GNN, the ant colony algorithm utilizes a pheromone update mechanism and heuristic strategy to optimize path selection in the graph gradually. Ant colony algorithm utilizes global search capability to adjust paths, strengthen the selection of high-quality paths, and continuously optimize the distribution of pheromones, making the GNN model more accurate and efficient in the path planning process, thereby achieving global optimization of paths.

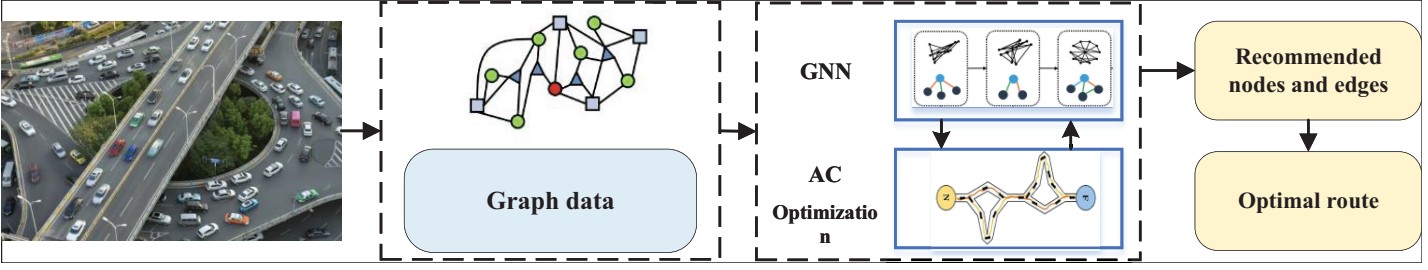

**Figure 2 The framework for optimal route planning.**           

# EXPERIMENT RESULT AND ANALYSIS

## Experiment setup and dataset description

In this article, the input trajectory data is compared with the final formation of the actual optimal trajectory to complete the corresponding binary classification task for analysis (https://zenodo.org/records/3243562, doi: 10.5281/zenodo.3243562). If the prediction of the nodes and the edited nodes are consistent, the prediction is considered accurate; otherwise, it is deemed inaccurate. Node and edge prediction results are utilized to evaluate the model's effectiveness. Given that the GNN classification model addresses the trajectory-based path planning problem and encounters significant category imbalance, this article employs extended recall, extended precision, and extended F1-score metrics for evaluation (*Battaglia et al., 2018*).

$$EX\_\text{precision} = \frac{TPN + TPE}{(TPN + FPN) + (TPE + FPE)} \tag{9}$$

$$EX\text{ recall} = \frac{TPN + TPE}{(TPN + FNN) + (TPE + FNE)} \tag{10}$$

$$EX\_F1\text{-score} = \frac{2 \cdot EX\_\text{recall} \cdot EX\_precision}{EX\_\text{recall} + EX\_precision} \tag{11}$$

TPN (true positive on nodes) denotes actual examples in node results, FNN (false negative on nodes) denotes false negative examples in node results, and FPN (false positive on nodes) denotes false positive examples in node results. TPE (true positive on edges) denotes actual cases on edge results, FNE (false negative on edges) denotes false negative cases on edge results, and FPE (false positive on edges) denotes false positive cases on edge results. Extended recall and extended precision essentially compute the overlap between the actual trajectory path and the predicted path. Under the problem of class imbalance, traditional precision and recall tend to lean towards the majority class, resulting in insufficient performance evaluation of the minority class. The use of extended recall rate, extended precision rate, and extended F1 value can more effectively measure the prediction quality of minority classes, making the evaluation indicators fairer in the case of class imbalance, ensuring that the model can focus on the performance of minority classes when

**Table 1 The experiment environment information.**

| Environment | Information |
|---|---|
| CPU | I7-14700F |
| GPUs | RTX 3060Ti |
| Language | Python 3.5 |
| Framework | TensorFlow |

dealing with trajectory-based path planning problems. These extended indicators enhance the model's reliability in real-world applications, providing a more accurate and balanced evaluation method for path planning. Additionally, given the involvement of a deep learning model, we constructed the experimental environment with specific parameters detailed in Table 1.

For the selection of public datasets, this article utilizes the publicly available vehicle trajectories from DiDi (*Wang et al., 2020*) for model testing under public datasets. Additionally, we selected route shortest optimization and deep learning algorithms commonly used in the field for the comparison model. The main comparison methods include Dijkstra's shortest path algorithm (*Chen, 2022*), random graph network (RGN) (*Yu, Guo & Chen, 2020*), and the GNN algorithm without ant colony optimization for data analysis. Dijkstra's shortest path algorithm serves as the traditional graph theory algorithm baseline.

## Method comparison and result analysis

After constructing the model and confirming the dataset, we proceeded with the training and analysis of the model using the public dataset. The training activity and the corresponding changes in the indicators under the public dataset are illustrated in Fig. 3:

In Fig. 3, we observe that due to the model optimization performed *via* the ant colony algorithm for the GNN parameters after each iteration, the proposed ACGNN method exhibits superior and faster convergence performance. Compared to traditional shortest-path planning, better results are achieved through the graph network. The main computational indexes are presented in Fig. 4.

Figure 4 shows that under metrics such as Ex_precision, the ACGNN method proposed in this article performs significantly better than methods like RGCN and GNN. While it shows only a 15% improvement over the traditional shortest path method under the three metrics, the results are still noteworthy. Besides these quantitative indicators, we also analyzed and calculated the training time of the model. The time distribution table in Fig. 4 reveals that although ACGNN is not optimal under the public dataset, its overall time consumption remains acceptable for path planning research requiring high accuracy and stringent standards. In addition to analyzing the public dataset, we also conducted further study using a self-built dataset, which will be discussed in detail in the following subsection.

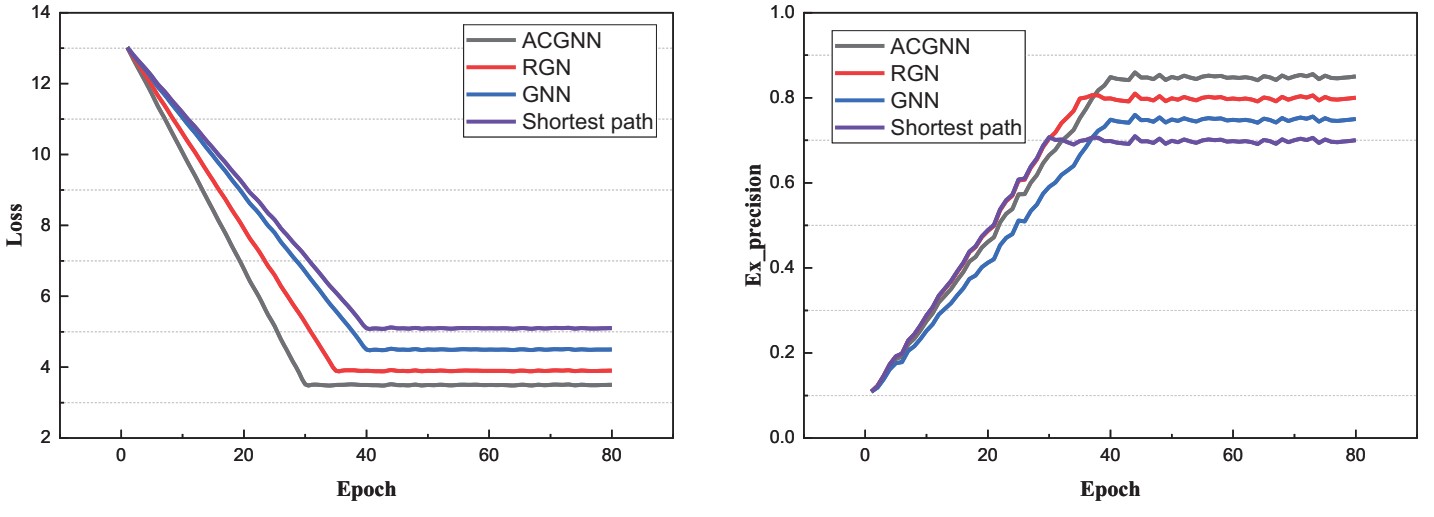

**Figure 3** The training process on the public datasets.               

## Model deployment and test

After completing the data collection under the public dataset, we further analyzed the data from this region, which still originates from the public database of DiDi. We selected path information from different periods within five days in this region as the test data, collecting over 5,000 valid data points. This data was used to complete the model training and the corresponding analysis. The results of our tests under this dataset are shown in Fig. 5.

In Fig. 5, the results are presented based on the training dataset. We opted not to use the usual 7:3 or 8:2 ratio for the training process but instead divided the data by a 5:5 ratio to address the imbalance in the overall sample data. The 5:5 split was chosen to ensure that training and validation sets had a representative distribution of different sample types. By doing so, we aimed to create a more balanced evaluation framework that would more accurately reflect the model's performance, preventing any one class from disproportionately influencing the results during training and helping to mitigate overfitting. The results indicate that the ACGNN method employed exhibits superior and faster path planning performance. The overall edge and node fitting performance better aligns with the optimal path repeatability, achieving an Ex-precision of over 0.85. Furthermore, the ACGNN method outperforms traditional methods in the other two indicators. Regarding running time comparison, the ACGNN method operates within seconds, suggesting that smaller data samples can achieve excellent path-planning capabilities with lower computational resource consumption. To further test the optimization capability of ACO algorithms for models in path planning studies, we conducted a comparative analysis of several standard meta-heuristic optimization algorithms. The results of this analysis are shown in Fig. 6.

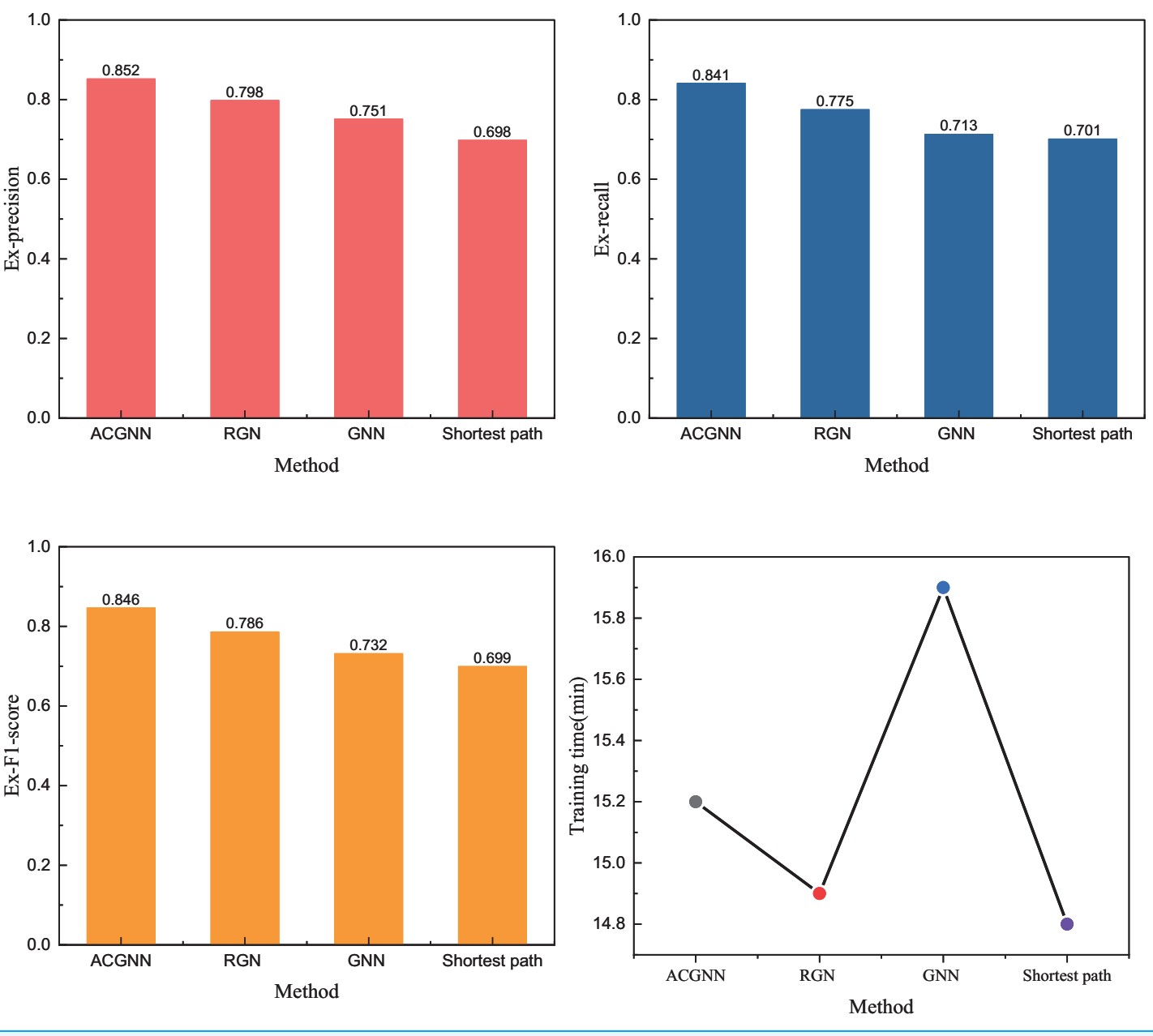

**Figure 4 The experiment results on public datasets.**

In comparing meta-inspired algorithms, we selected two widely used and classical algorithms, simulated annealing and PSO, for comparison. As shown in Fig. 6, compared to a single GNN method, the overall path planning performance is enhanced by optimizing meta-heuristic algorithms. Among these, the ACO algorithm demonstrates superior optimization performance over the other two algorithms. While the PSO method performs better under the precision index, the ACO algorithm achieves the best overall optimization results.

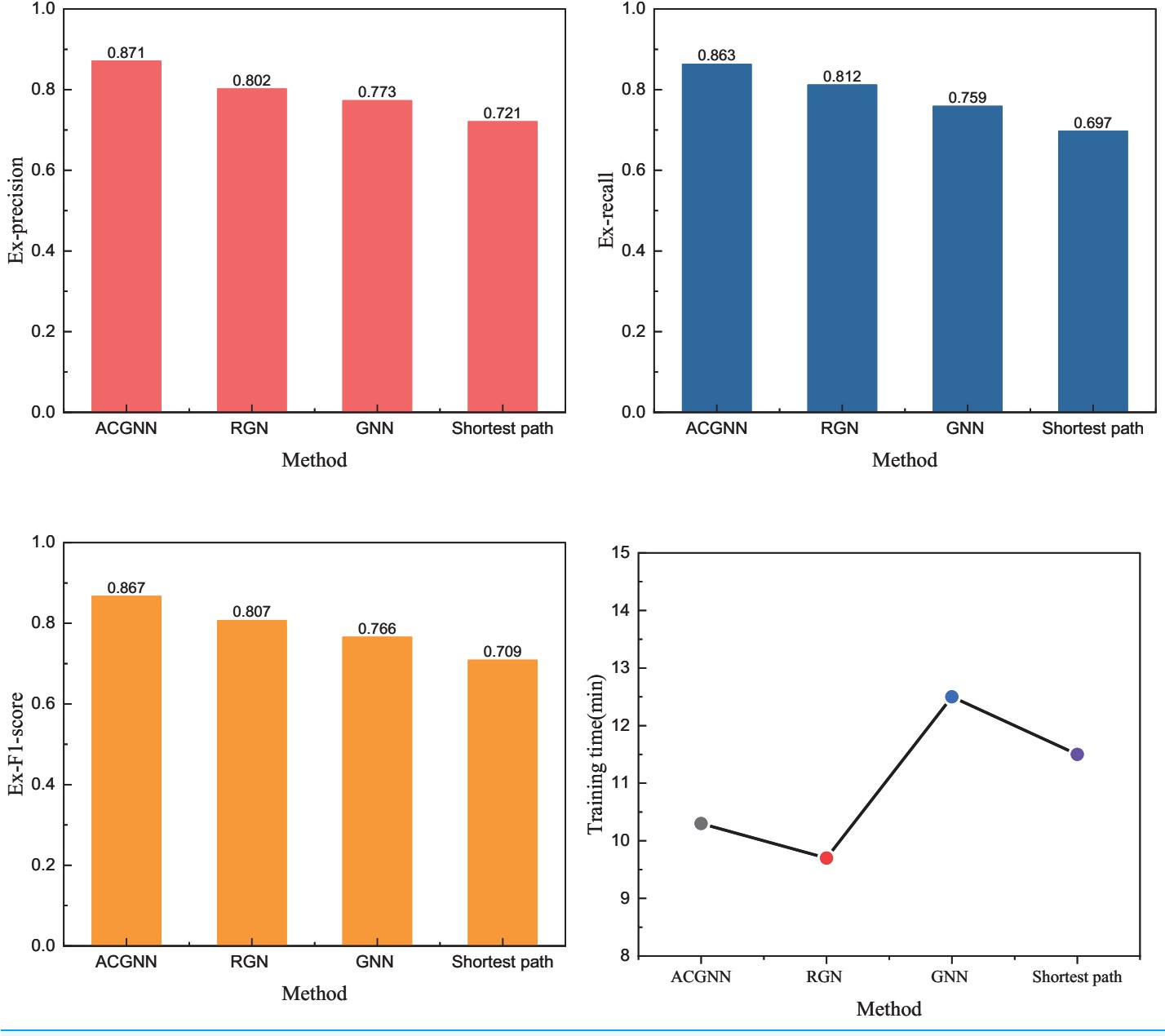

**Figure 5 The experiment results on self-established datasets.**

We examined the computation time of different meta-inspired optimization algorithms and various optimization steps to analyze the optimization performance further. The comprehensive results of this analysis are displayed in Fig. 7.

Regarding overall model running time, models optimized through meta-heuristic algorithms achieve faster convergence speeds and a significant performance improvement. The ant colony algorithm exhibits the best optimization effect and the quickest speed. Additionally, we varied the number of optimizations during the training process, setting it

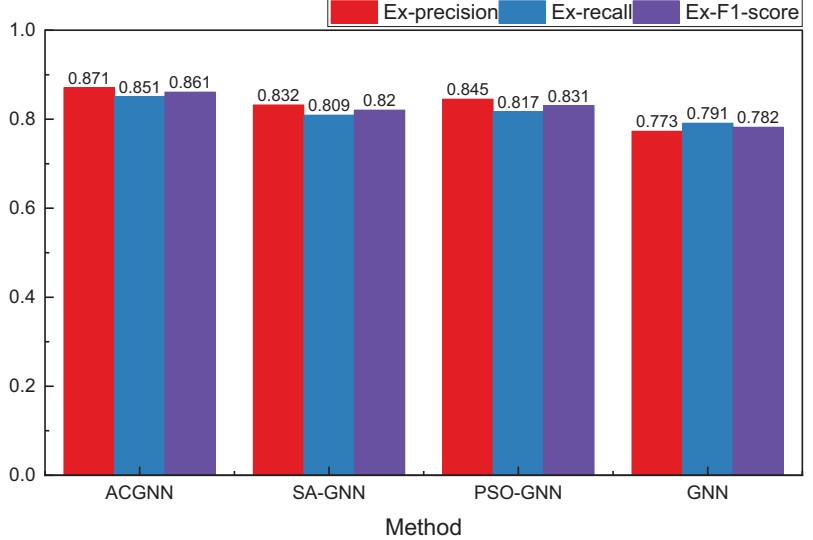

**Figure 6  The comparison results concerning different optimization method.**

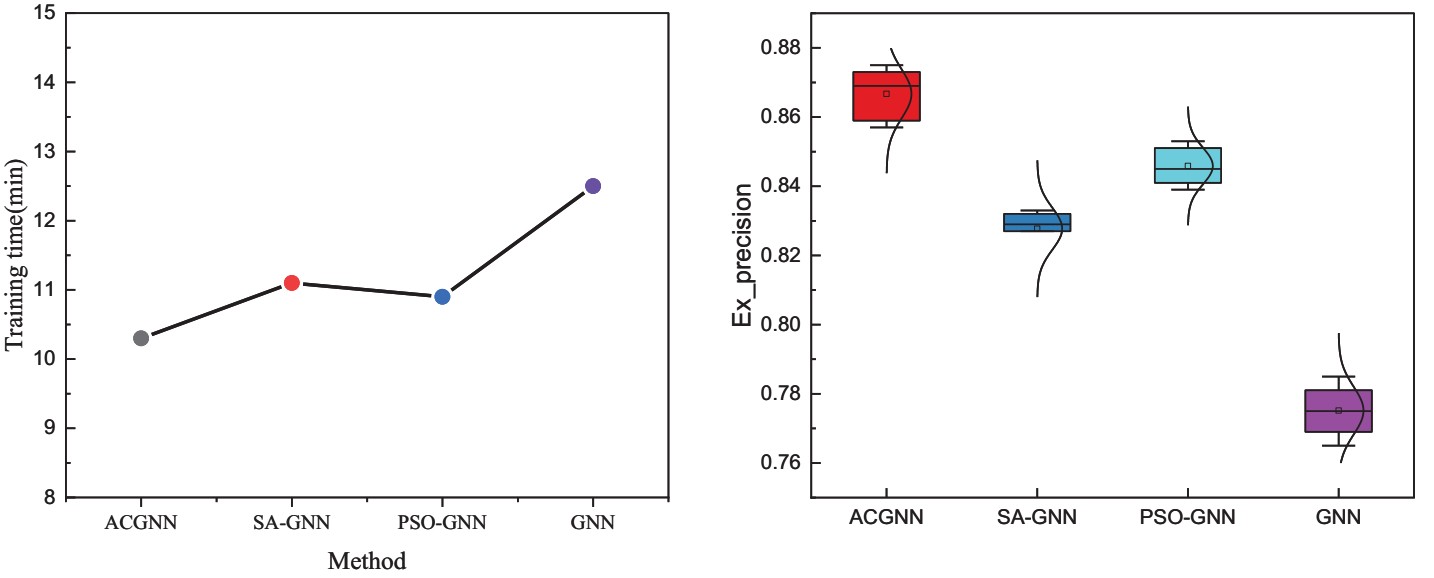

**Figure 7  Comparison of results across various optimization methods.**

to 5, 10, 15, up to 30 times, and updated every five times to observe the final results. The results, illustrated in the boxplot, show that the ant colony algorithm demonstrates superior performance in updating, consistently outperforming other methods at different time steps. The ACGNN model has broad potential value in practical application scenarios, especially in intelligent transportation systems. By combining path optimization and real-time decision-making, ACGNN can analyze dynamic data in transportation networks, provide optimal route planning, help alleviate congestion, improve road traffic

efficiency, and achieve efficient vehicle scheduling and management. In intelligent transportation applications, deploying ACGNN models can significantly improve fuel efficiency and reduce idle time for vehicles on congested roads, thereby reducing carbon emissions and achieving energy-saving and emission-reduction goals. In addition, through precise control of path planning, this model can optimize the scheduling of emergency vehicles, improve accident response speed, and ensure the safety and smoothness of urban traffic. Overall, the application of ACGNN in intelligent transportation improves travel efficiency and has high environmental and social benefits, which profoundly impact the future construction of smart cities.

## DISCUSSION

This article analyzes the optimal route planning problem in unmanned and autonomous driving and related navigation path planning research, proposing a particle swarm optimization-based ACGNN framework. This framework ensures the diversity and accuracy of the input data by utilizing graph data generated from road networks and historical trajectories as model inputs, along with generative graph data and subgraph filtering techniques. Compared to traditional path planning methods, ACGNN optimizes the computation of nodes and edges through GNN, allowing the model to capture complex road network structures and dynamically changing traffic information. Furthermore, introducing the ACO algorithm provides a robust optimization mechanism for GNN. With its excellent global search ability and adaptability, ACO efficiently explores and reinforces the optimal path, avoiding local optima by simulating the ants' foraging behavior and leveraging the positive feedback mechanism of pheromones. This results in superior performance and stability compared to SA and PSO in complex path planning problems, significantly enhancing the GNN's node and edge computation optimization performance. Experimental comparisons on public and self-constructed datasets demonstrate that ACGNN outperforms Dijkstra's shortest path algorithm, RGN, and traditional GNN methods in path planning, highlighting its superiority in path optimization.

Path planning is a crucial technical tool for achieving efficient logistics and traffic management in modern transportation systems. The ACGNN method introduces a more intelligent and precise computational model for path planning by integrating generative graph data and GNN. While effective in specific environments, traditional path planning methods, such as Dijkstra's shortest path algorithm, RGN, and traditional GNN approaches, often fall short when confronted with complex and dynamic road networks and traffic conditions. Conversely, ACGNN significantly enhances the accuracy and efficiency of path planning by using graph data generated from road networks and historical trajectories as inputs, creating high-quality graph data through subgraph filtering techniques, and then optimizing the computation of nodes and edges with GNN. Compared to SA and PSO algorithms, the ACO algorithm performs better in avoiding local optima, making it particularly suitable for complex and variable path optimization problems. With increasingly stringent global requirements for carbon emission control and environmental protection, the transportation industry is under significant pressure to

reduce emissions. ACGNN can effectively minimize vehicle mileage and time through intelligent path optimization, thus lowering fuel consumption and carbon emissions. Additionally, optimized path planning can alleviate traffic congestion, improve transportation efficiency, and reduce emissions caused by vehicle idling and frequent stops. Therefore, beyond providing crucial decisions for optimizing traffic flow allocation and efficient urban traffic management, ACGNN holds substantial significance for the low-carbon and green development of entire cities.

## CONCLUSION

The vehicle path low-carbon planning model ACGNN based on an adaptive multi-strategy ant colony optimization algorithm proposed in this study offers an effective solution to the path planning problem in complex traffic environments. By integrating GNN and the ant colony optimization algorithm and incorporating a strategy for generating graph data, an optimization system capable of processing and analyzing the characteristics of complex road networks is constructed. This system successfully realizes efficient low-carbon path planning. The experimental results confirm the superior performance of the model in practical applications, with its path optimization effect and other evaluation metrics significantly surpassing those of traditional methods such as Dijkstra's shortest path algorithm, RGN, and conventional GNN methods. In tests on public and self-built datasets, ACGNN achieves notable improvements in path planning accuracy, demonstrating clear advantages over traditional methods in various optimization scenarios. The model's accuracy improves by 15% compared to the basic shortest path algorithm, showcasing its potential application in intelligent transportation systems. This research advances the development of intelligent transportation technology and provides new technical means and strategic support for the design and low-carbon optimization of transportation systems.

In our future research, we aim to broaden the application of our model by integrating a more decadent array of traffic scenario data, including high-density urban areas, varying traffic flow conditions, and unique road types such as mountainous and rural routes. This will allow the model to account for a broader range of real-world situations, thus enhancing its robustness and utility across diverse environments. Additionally, we plan to incorporate advanced AI methodologies, such as deep reinforcement learning for real-time decision-making in unpredictable traffic, and evolutionary algorithms to optimize model parameters dynamically in response to changing road conditions.

To improve adaptability, we will focus on developing a hybrid approach that combines our current framework with CNNs to capture spatial patterns in traffic flow and integrate GANs to simulate potential traffic scenarios for enhanced model training. Refining the model's architecture and introducing self-learning mechanisms, we aim to support more complex and scalable path-planning solutions prioritizing low-carbon objectives. Ultimately, these advancements will provide sustainable, efficient, and adaptive technical support for intelligent transportation systems, advancing environmental and operational goals.

### Funding
This work is funded by "Research on Coupling Optimization of Urban Underground and Surface Logistics Distribution Network Based on Subway", the project number is 51808187; and also funded by "Site Selection and Route Optimization of China-Europe Express in Jiangsu Province—Jiangsu Provincial Natural Science Foundation (Youth Project)", project number: BK20170879. The funders had no role in study design, data collection and analysis, decision to publish, or preparation of the manuscript.

### Grant Disclosures
The following grant information was disclosed by the authors:
Research on Coupling Optimization of Urban Underground and Surface Logistics Distribution Network Based on Subway: 51808187.
Site Selection and Route Optimization of China-Europe Express in Jiangsu Province—Jiangsu Provincial Natural Science Foundation (Youth Project): BK20170879.

### Competing Interests
The authors declare that they have no competing interests.

### Author Contributions
- Qi Guo performed the experiments, analyzed the data, prepared figures and/or tables, and approved the final draft.
- Rui Li conceived and designed the experiments, analyzed the data, prepared figures and/or tables, and approved the final draft.
- Changjiang Zheng conceived and designed the experiments, performed the experiments, performed the computation work, authored or reviewed drafts of the article, and approved the final draft.
- Gwanggil Jeon conceived and designed the experiments, performed the experiments, prepared figures and/or tables, authored or reviewed drafts of the article, and approved the final draft.

### Data Availability
The data is available at Zenodo: Li, Wan; Guo, Qiangqiang; Ban, Xuegang. (2019). Vehicle trajectory data in simulation network [Data set]. Zenodo. https://doi.org/10.5281/zenodo.3243562.
The original source of the data:
- didi chuxing. http://www.didichuxing.com.
- https://dl.acm.org/doi/abs/10.14778/3384345.3384348.

### Supplemental Information
Supplemental information for this article can be found online at http://dx.doi.org/10.7717/peerj-cs.2611#supplemental-information.

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
