# Peer review of "Design of low-carbon planning model for vehicle path based on adaptive multi-strategy ant colony optimization algorithm"

_PeerJ Computer Science, doi:10.7717/peerj-cs.2611_

## Round 0.1 · original submission · Major Revisions

Dear authors,

Thank you for the submission. The reviewers’ comments are now available. It is not suggested that your article be published in its current format. We do, however, advise you to revise the paper in light of the reviewers’ comments and concerns before resubmitting it. The followings should also be addressed:

1. Information about the datasets should be provided in the Abstract section.
2. Pay special attention to the usage of abbreviations. Spell out the full term at its first mention, indicate its abbreviation in parenthesis and use the abbreviation from then on.
3. The motivation and reason of using multi-strategy ant colony optimization algorithm among many other metaheuristic algorithms for feature selection should be mentioned. Furthermore, the reason about comparing the results only with simulated annealing algorithm and particle swarm optimization algorithm should be provided.
4. Equations should be used with correct equation number. Many of the equations are part of the related sentences. Attention is needed for correct sentence formation.
5. All of the values for the parameters of all algorithms should be given.
6. Encoding type or representation scheme and fitness function of colony optimization algorithm should be provided. How constraints for metaheuristic methods (for example in decision variables intervals) are handled is not clear.

Best wishes,

Reviewer 1 ·

Basic reporting

These reviews focus on improving the depth of your technical content and methodology. I hope these suggestions are helpful as you work toward enhancing your manuscript.

(1) On Integration of Metaheuristic Algorithms (p.9, Section 3.1): The manuscript introduces the combination of the ant colony algorithm with GNN for path planning, but the description of how these two work together is too brief. Please provide more details on how they are integrated at the algorithmic level and how the ant colony algorithm specifically enhances GNN performance.

(2) On Evaluation Metrics (p.13, Section 4.2): The metrics used in the experiments, such as Ex-precision, are not well-explained. I recommend further clarification of the meaning of these metrics and how they differ from traditional metrics (e.g., precision, recall), along with an explanation of their significance to the field.

(3) On Real-World Application Scenarios (p.17, Conclusion): While ACGNN shows promise in theory, the manuscript lacks discussion of real-world applications. Please include an analysis of practical application scenarios, such as how this model can be deployed in intelligent transportation systems, and assess its potential impact in terms of benefits like fuel efficiency or reduced emissions.

Experimental design

(4) On the Complexity of Path Planning Problems (p.3, Introduction): The introduction highlights that “path planning is a critical challenge,” but there is no in-depth exploration of the complexity of path planning problems. I suggest adding a discussion on factors such as the dynamic nature of urban traffic networks, real-time data processing, and environmental uncertainties to provide a more comprehensive view of the problem.

(5) On Environmental Impact (p.3, Introduction): The manuscript stresses the importance of low-carbon path planning, but there is no concrete experimental or theoretical support. I recommend including a quantitative analysis of how the path optimization reduces carbon emissions, thus reinforcing the environmental significance of the research.

(6) On Parameter Tuning of Ant Colony Algorithm (p.9, Section 3.1): The parameters of the ant colony algorithm, such as the importance of pheromone concentration and volatility, are not discussed. A sensitivity analysis of these parameters would provide insights into how they influence the model’s performance.

Validity of the findings

(7) On GNN’s Computational Bottleneck (p.10, Section 3.2): While GNN is effective at processing graph-structured data, it may face computational bottlenecks with larger graphs. Please discuss the model’s performance on larger graph datasets and consider suggesting optimization strategies to handle large-scale graphs.

(8) On Future Research Directions (p.17, Conclusion): The conclusion would benefit from a more detailed discussion of future research directions, particularly integrating advanced AI techniques such as reinforcement learning or evolutionary algorithms to improve adaptability and robustness in path planning tasks.

Reviewer 2 ·

Basic reporting

By defining the different kinds of cars or the facets of complexity (such as energy usage and real-time traffic data), the statement "burgeoning number and complexity of vehicles" might be clarified.

The sentence "Amalgamating advanced information technology..." needs to be shorter. Think about making it simpler to read "Integrating advanced IT systems..."

The expression "enhancing optimization efficiency and effectiveness" is used a lot. Select "effectiveness" or "efficiency" to make the phrase more concise.

The reference diagram In the text, "Figure 2" is not obvious. Before mentioning the figure, provide a phrase that explains what it depicts.

When you use the term "due to the model optimization performed via the ant colony algorithm," think about being more precise about the type of optimizations that were carried out.

Experimental design

The 5:5 data split explanation seems inadequate. Explain the rationale behind using this ratio instead of the more common 7:3 or 8:2 split.

Validity of the findings

The statement "ACGNN holds substantial significance for the low-carbon and green development" needs to be supported by more specific instances of its importance, such as efficiency improvements or emissions reductions.

---

## Round 0.2 · accepted · Accept

Dear Authors,

Thank you for addressing the reviewers" comments. Your manuscript now seems ready for publication.

Best wishes,

Reviewer 1 ·

Basic reporting

All changes have been completed.

Experimental design

All changes have been completed.

Validity of the findings

All changes have been completed.

Additional comments

All changes have been completed.

Reviewer 2 ·

Basic reporting

Thank you for submitting the revised version of your manuscript. I have carefully reviewed the changes you have made in response to my previous comments. You have addressed all of my concerns in a satisfactory manner. The manuscript is now of a high quality and I recommend it for publication.

Experimental design

Thank you for submitting the revised version of your manuscript. I have carefully reviewed the changes you have made in response to my previous comments. You have addressed all of my concerns in a satisfactory manner. The manuscript is now of a high quality and I recommend it for publication.

Validity of the findings

Thank you for submitting the revised version of your manuscript. I have carefully reviewed the changes you have made in response to my previous comments. You have addressed all of my concerns in a satisfactory manner. The manuscript is now of a high quality and I recommend it for publication.